

**Māori oral histories and the recurring impact of tsunamis in Aotearoa-New**
**Zealand**
Darren N King[1][2], Wendy S Shaw[2], Peter N Meihana[3], James R Goff[2]
1. Māori Environmental Research Centre – Te Kūwaha o Taihoro Nukurangi, National Institute of

6         Water and Atmospheric Research Ltd (NIWA), Aotearoa-New Zealand.

2. PANGEA Research Centre, School of Biological, Earth and Environmental Sciences, University

8         of New South Wales (UNSW), Australia

3. School of Humanities, Massey University, Aotearoa-New Zealand.
Author correspondence
Phone: +64 9 3752086
Email: darren.king@niwa.co.nz
Address: Māori Environmental Research Centre – Te Kūwaha o Taihoro Nukurangi, National Institute
of Water and Atmospheric Research Ltd, Private Bag 99940, Auckland, Aotearoa-New Zealand.
Key words: Tsunami, Palaeotsunami, Oral history, Māori, Aotearoa, New Zealand



ABSTRACT
Māori oral histories from the northern South Island of Aotearoa-New Zealand provide details of
ancestral experience with tsunamis. Exchanges with key informants from the Māori kin groups of
Ngāti Koata and Ngāti Kuia reveal that these histories, recorded in a narrative form, are not merely
another source of information about past catastrophic saltwater inundations but, rather, reference
multiple layers of experience and meaning, from memorials to ancestral figures and their
accomplishments, to claims about place, authority and knowledge. Notwithstanding these
confirmations, to engage as insider-outsiders with Māori oral histories (and the people who
genealogically link to such stories) requires close attention to a politics of representation as well as
sensitivities to the production of 'new' and 'plural' knowledge itself. Individuals and families from
Ngāti Koata and Ngāti Kuia permitted us to record some of *their* histories. They share the view that
there are multiple benefits to be gained by learning from differences in knowledge, practice and
belief. This paper makes these narratives available to a new audience (including those families who no
longer have access) and recites these in ways that might encourage those more intimately connected
to know and transmit these histories differently.
WHAKARĀPOPOTOTANGA
Ko ngā kōrero tuku Māori o Te Tauihu o te Waka a Māui e whakaahua nei i tā ngā tūpuna rongo i te
aituā nui o te parawhenua waitai. Nā runga i ētahi whakawhitinga kōrero ki ētahi māngai matua o
Ngāti Koata me Ngāti Kuia, i mārama ai ko ēnei kōrero tuku, he mea mau ā-pakiwaitara nei, ehara noa
i te puna kōrero mō te tai āniwhaniwha o nehe, engari kē, he mea whai tikanga maha, mai i te
whakamaumahara i ētahi tūpuna o nehe me ngā mahi i oti i a rātou, tae atu ki ngā kōrero mō te rohe,
mō te mana, mō te mātauranga anō. Hāunga ēnei whakaūnga, e whai kiko ai te whai wāhi atu hei
'rāwaho-whai-hononga' ki ngā kōrero tuku Māori (me te hau kāinga e hono ā-whakapapa ana ki ngā
kōrero), me aro pū ki te taha tōrangapū o te tū hei māngai mō iwi kē, ā, me ngā kaupapa mana nui me
mātua whakaaro i te whakaritenga o te mātauranga 'hōu', o te mātauranga 'mātāpuna-tini' anō. I
whakaae mai ētahi māngai takitahi me ētahi whānau anō o Ngāti Koata me Ngāti Kuia kia hopukina





ētahi o *ā rātou* kōrero tuku. E whakaae ana rātou he hua nui ka puta i te whai māramatanga ki ngā
rerekētanga ā-mātauranga, ā-tikanga, ā-whakapono anō. Ko tā ngā kōrero i tēnei tuhinga he
whakawātea i ngā pakiwaitara tuku nei ki tētahi whakaminenga hōu (tae atu ki ngā whānau kāore i
whai wāhi ki ngā kōrero nei i mua), ā, ko te āhua e takoto nei ēnei kōrero hei akiaki pea i ērā e whai
hononga ana kia mātau ka tahi, ka rua, kia tuku hoki i ngā kōrero mā ara kē atu anō.



## 1. INTRODUCTION

"What is all this? " he asked. "These are the fish I have caught," replied Titipa. "This is the result of my power as a *tōhunga* [priest; expert in traditional lore; person skilled in specific activity; healer]." "But didn't I tell you I should expect the pick of the catch?" cried Te Pou. "If you want fish, catch them yourself," retorted Titipa. "You don't get the pick of my haul." "Indeed," said Te Pou, and he walked along the beach and inspected the fish that were drying in the sun. "We shall see whose catch this is presently." Walking to the water's edge and stretching out his arms towards the sea, he repeated mighty spells before the people. Everyone wondered what would happen, but it was not long before Te Pou came running up the beach. "Get back!" he cried. "Get back to the high ground, or you will be drowned," and running past his people he climbed the high cliff, where he took his stand, and repeated more spells. The people, thoroughly terrified, followed helter-skelter, and left Titipa alone upon the beach. Soon the sea grew dark and troubled and angry, and presently a great wave, which gathered strength as it came, swept towards the shore. It advanced over the sandy beach, sweeping Titipa and all his fish before it, till with the noise of thunder it struck the cliff on which the people stood. "That is one," said Te Pou. "That is for the first fish. There will be two more." The great wave receded, sucking with it innumerable boulders and the helpless, struggling Titipa. Then another wave, greater than the previous one, came with tremendous force and, sweeping the shore, struck the cliff with a thunderous roar. This was followed by a third which, when it receded, left the beach scoured and bare. Titipa and all his fish had disappeared. "I have finished," said Te Pou. "That is all. There will be no more trouble…"

[The Rival Wizards: Grace, 1907a]

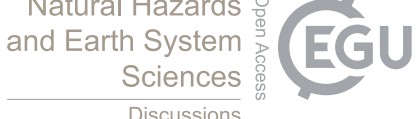

In 1907, Alfred Grace (1867-1942) published a series of Māori "folk stories", imparted by the Ngāti
Koata[1] elder Karepa Te Whetu. Within the extensive narrative of one of these stories, 'The Rival
Wizards' the "wizard-chief", Te Pou, summoned three great waves to exact retribution upon the rival
Titipa for openly defying his instructions. Descriptive details of the impact of great waves striking and
scouring the beach were narrated, including many contextual details about the relationships and
connections between people, place and the metaphysical world. The reciting of this narrative in print,
however, did not occur again until King et al. (2007) and McFadgen (2007) cited the story, among
other traditional stories, and made a case for the scientific value of Māori oral histories in
understanding catastrophic saltwater inundations or tsunamis in pre-colonial Aotearoa-New Zealand
(A-NZ). King and Goff (2010) surmised that the descriptive nature of the language in the story
resembled those of modern-day tsunami survivors and argued that it might represent an historical
narrative recording direct experience with one (or multiple) tsunami inundations, prior to the arrival
of the first Europeans to A-NZ in the late eighteenth century. However, they also acknowledged that
the interpretation of Māori stories by 'outsiders' is fraught with the potential for misrepresentation
and concluded the need to engage with Māori who share ancestral and kinship linkages with specific
oral histories to tell our/their own stories.

---

[1] Ngāti Koata is one of several Māori kin-groups [*iwi*] who hold territorial rights, power and authority

associated with possession and occupation of *iwi*-land over the northern South Island (Mitchell and

Mitchell, 2004). They date their occupation in the area from the late 1800's, and recognise the

successive movements of earlier peoples migrating to and through the area. Details surrounding

occupational patterns are provided in: Keyes (1960), Mitchell and Mitchell (2004).



This study builds upon these collective contributions by working alongside key informants from the
Māori kin groups of Ngāti Koata and Ngāti Kuia[2] from the northern coast of the South Island of A-NZ
(Figure 1). These informants share linkages not only with Karepa Te Whetu but also the places and
ancestral figures named in the 'The Rival Wizards' story. The paper begins by providing an overview of
past work in the geosciences to have benefitted from the insights provided by indigenous oral
histories. This necessarily includes a brief review of complementary lessons in political,
epistemological and methodological theory. The research framing for this work and the methods of
analysis are next outlined, before providing detailed accounts of the key elements of the story
supported by examples of contemporary dialogue, discussion and conversation. Finally, consideration
is given to the lessons, challenges and opportunities that can come from bringing the knowledge-
practice-belief complex of Māori Knowledge [Mātauranga Māori] together with the earth system
sciences.
**2. INDIGENOUS ORAL HISTORIES AND TSUNAMIS**
Consideration of Indigenous oral histories as tsunami narratives is not new. Vitaliano (1973) discussed
the scientific benefits to be gained by considering "myths and legends" as transmission devices for
knowledge about (and experience with) tsunamis, among other geologic phenomena. Her work
detailed examples of coastal deluge attributed to tsunamis (and their likely sources) from classical
Greek history through to more recent times from the Pacific coasts of the Americas to islands across
the Pacific Ocean. Accordingly, Vitaliano (1973) argued that such insights provide invaluable
information about extreme environmental disturbances in the pre-written past. A series of scientific
contributions have since emerged from the Pacific Northwest coast of North America detailing 'Indian
myths' and the transmission of knowledge about great sea level disturbances (Heaton and Snavely,

---

[2] Ngāti Kuia is one of several Māori kin-groups (*iwi*) who hold territorial rights, power and authority

associated with possession and occupation of *iwi*-land over the northern South Island. They are often

referred to as one of the ancestral iwi of the region (Mitchell and Mitchell, 2004).



1985; Clague, 1995; Hutchinson and McMillan, 1997; McMillan and Hutchinson, 2002; Ludwin et al.,
2005; Ludwin and Smits, 2007; Thrush and Ludwin, 2007; Vitaliano, 2007).
Heaton and Snavely (1985) and Clague (1995) concluded that many details within indigenous oral
histories are consistent with tsunami inundation processes (e.g. the sudden receding of coastal
waters). Recognising this experience with earthquakes and tsunamis along the northern Washington
and southern British Columbia coasts McMillan and Hutchinson (2002) argued that oral histories can
provide independent sources of information which can complement geological and archaeological
knowledge about the role of infrequent yet catastrophic events in landscape evolution and social-
cultural transformation. They also made explicit that such histories may have other independent
meanings. Advancing this scholarship, Ludwin et al. (2005) considered 40 stories from 32 independent
sources about coastal earthquakes and marine flooding; and with help from Japanese historical
records determined that the most recent large-scale event captured in multiple stories along the
Cascadia coast occurred on 26 January 1700. Importantly, Thrush and Ludwin (2007) recognised that,
Native American and First Nations oral histories not only include rich and explicit accounts of seismic
events, but also that scientific inquiry is grounded in the historical relationships between indigenous
and settler societies, and that this has resulted in the privileging and production of certain kinds of
knowledge about the region's seismic past. Likely informed by transformative and decolonising
research theories, this corollary point raised important questions about geology's relationship with
colonialism, intellectual and cultural property, as well as the complex and fractious relationships
between researchers and the researched. Thrush and Ludwin (2007) highlighted the tremendous
potential for benefitting from differences in knowledge, practice and belief about some of the largest
seismic events known to human-kind.
Considerable scholarship has outlined the scientific value of indigenous expertise and information
about tsunamis referenced in oral histories from the Pacific Islands (Nunn, 2001; Lum-Ho and Lum-
Ho, 2005; Nunn and Pastorizo, 2007; Goff et al., 2008; Stewart, 2009; Goff et al., 2011; Johnston and
Dudley, 2009) and in A-NZ (Goff et al., 2003; King et al., 2007; McFadgen, 2007; McFadgen and Goff,



2007; King et al., 2010; Pearce and Pearce, 2010; Goff et al., 2012; Goff and Chagué-Goff, 2015; King,
2015; King et al., 2017). Further, there are likely to be contributions from other non-English science
communities about the potential value of indigenous histories enriching the geo-archaeological
sciences, but such references were not identified in the sweep of English language scholarship
conducted here. Notable contributions from the Pacific include Nunn (2001), who identified
ethnographic narratives of probable experiences with tsunami inundation, including a story from
Pukapuka Atoll in the northern Cook Islands where time is divided into before and after a huge wave
swept over the island. Nunn and Pastorizo (2007) also identified that Pacific Islander 'myths' might
inform the chronology and social impacts of such hazards. Similarly, Hawaiian scholars are also re-
examining their own oral histories that relate an extended history of exposure to tectonic and
geologic hazards – including tsunamis (Lum-Ho and Lum-Ho, 2005; Stewart, 2009). This work is as
much about adding to the scientific pool of scholarship surrounding Hawaii's tsunami risk-scape as it
is about cultural revitalisation and connecting with the ancestors.
Meanwhile in A-NZ, Goff et al. (2003) emphasised the limited time frame of the historical record for
understanding tsunami risk in A-NZ and thereby pointed to the Māori oral record as a potentially rich
source of information about tsunamis occurring prior to European arrival. Succeeding this work, there
have been varying attempts to link geo-archaeological evidence and modelling output with historical
events inferred from Māori tsunami narratives (Walters et al., 2006; McFadgen and Goff, 2007; King
and Goff, 2010). King et al. (2007) argued that Mātauranga Māori is a neglected area of expertise in
scientific assessment and declared that greater Māori involvement is required in natural hazards
science to make the most of all the knowledge and skills that Māori possess. After this, King and Goff
(2010) mapped selected Māori oral histories that potentially related experience with tsunamis around
the A-NZ coast. These narratives were compared with contemporary scientific data and the
implications of this 'new' information for tsunami science were considered. Importantly, this work
signalled the need for new research approaches that openly and respectfully engage with Māori who
hold ancestral and kinship linkages to oral histories to tell our/their own stories. Such perspectives





have the potential to amend (and perhaps replace) accepted scientific views about pre-colonial
tsunami disturbance and risk in A-NZ.
**3. DEVELOPMENTS IN POLITICAL, EPISTEMOLOGICAL AND METHODOLOGICAL THEORY**
Developments in political, epistemological and methodological theory from a range of disciplines are
relevant to research that explores the potential of indigenous narratives to inform about
environmental histories and extreme disturbances such as tsunamis. A key debate relates to how
knowledge is constructed and legitimised, including whether a meaningful transfer of knowledge
between different knowledge histories can occur (or alternatively do harm) when removed from its
cultural context. As Mikaere (1995) argued, the outcomes of early 'research on' Māori (or rather the
inaccurate recordings and imaginary portrayals of narratives) rendered oral histories as "fantasy" and
resulted in "epistemological disarray". Bishop and Glynn (1999) contend that this reflected the
inadequacy of non-Māori to understand and accept the nature of Mātauranga Māori. Whatever the
case may be an ongoing challenge is to understand that narratives embedded within indigenous
knowledge systems provide more than alternative sources of information or even alternative
perspectives (Binney, 1987; Smith, 1999; Mead, 2003). Rather they have their own purposes, which
may include devices that help to establish meaning for discrete and repeated events through time
(Masse et al., 2007).
According to Cruickshank (1994), debates or understandings about knowledge construction are as
much about "epistemology" as they are about "authorship". She explains that for many Indigenous
peoples there is a reluctance to analyse and publicly explain the meanings of oral histories as this
takes away from the value and different messages that come from listening to repeated tellings from
family and extended kin, in place. This contrasts with a scholarly approach which encourages the
scrutiny of texts, and contends that by openly addressing conflicting interpretations, meanings can be
determined to enrich understanding. Many Indigenous commentators are thereby challenging
researchers within the academy of science to reframe how they construct and use knowledge. This
includes the treatment of Indigenous experience and knowledge as archaic and unchanging which





can, without consequence, be used by science to produce "authoritative" and "universal" insights
(Howitt and Suchet-Pearson, 2003; Shaw et al. 2006; Coombes et al. 2010). In response, Johnson et al.
(2016: 3) argue "scientists have to learn to see our own privilege, our own context, our own deep
colonizing. We have to learn to think anew - to think in ways that take seriously and actually respond
to information, understanding and knowledges as if difference confronts us with the possibility of
thinking differently".
The production of knowledge is deeply entwined with power relationships and who holds control and
authority over knowledge and its applications (Stephenson and Moller, 2009). This challenge is based
on the premise that power underpins the place of science in contemporary society, and that the
narrators of science (and history) ultimately hold power, whether knowingly or not (Johnson et al,
2016). Indigenous commentators (and others) have discussed legacies of extractive research practice,
whereby non-Indigenous researchers have treated the holders of Indigenous knowledge as if they
have no moral or legal rights to decide how it will be represented or used within the wider world.
Such practices have often resulted in leaving those studied disenfranchised from the knowledge they
have shared (Kovach, 2009). Indigenous scholars have thereby mounted a critique of the way history
has been told from the perspective of the colonisers – and this has resulted in debates over who gets
to frame and legitimise knowledge, whose voices are prominent in these discussions, and for whom
the writing is being done (Smith, 1999). A number of scholars have also challenged the notion of
including 'voices' in projects that aim to speak (or write) on behalf of 'others' (Howett and Suchet-
Pearson, 2003). For example, Coombes et al. (2014, 849) argue that "research that took the once-
radical step of 'giving voice' now patronizes and silences those whose voice is quite capable of self-
expression". While we recognise as researchers and authors the contradiction in the work completed
here, we acknowledge at the same time the collaborative basis of the research and the contribution
such grounded histories provide to scholarship.
In response to these histories and ethical challenges, all of which are taking place against a broader
background of indigenous self-determination and cultural affirmation, there is increasing recognition



of 'decolonising' and 'counter-colonial' research methodologies that seek to reframe and transform
the way research and knowledge is produced (Smith, 1999; Mead, 2003; Kovach, 2007). Key elements
of this discourse (although not limited to) include (i) valuing not only specific forms of Indigenous
knowledge but also the values underpinning such systems, (ii) recognising the authority of Indigenous
peoples to determine the rules for producing new knowledge, (iii) safeguarding the authenticity of
indigenous narratives, (iv) supporting research that enriches everyone who is connected with the
research project, and (v) promoting the benefits that come from learning from different ways of being
and knowing. Howitt and Suchet-Pearson (2003: 559) remind us also that "choosing whom to include
and how to include them, the choices other people have made in representing themselves to the
author and other authors, the ways the readers interpret the words and the ulterior motive for the
usage of the 'voices', all involve relationships of power".
**4. RESEARCH FRAMING**

233        **4.1 Methodological approaches**

This research applies an inductive-based methodological approach informed by 'collaborative
storytelling' to consider the meaning and memorials presented in the 'Rival Wizards' narrative. The
methodology does not fit neatly into any category, but draws on decolonising research approaches
(Smith, 1999; Kovach, 2009) and grounded theoretical principles (Glaser and Strauss, 1967; Pidgeon,
1996), while simultaneously seeking plural spaces of learning (Howitt and Suchet-Pearson, 2003;
Zanotti and Palomino-Schalsha, 2006; Johnson et al., 2016). This theoretical framing was underpinned
by Kaupapa Māori research principles (Smith, 1990; Te Awekotuku, 1991; Smith, 1999; Mead, 2003).
All informants were assured of their right to maintain authority over their contributions by reviewing,
editing and approving the 'new' narrative produced through this work. The National Institute of
Water and Atmospheric Research (HREC2017-005) and the University of New South Wales (HREC-
17085) provided human research ethics approvals.

245        **4.2 Methods, analysis and interpretation**



Semi-directive individual and paired interviews with 20 key informants from Ngāti Koata and Ngāti
Kuia were used to discuss the construction, key elements and purposes of 'The Rival Wizards'
narrative. In advance of all interviews a copy of the 'Rival Wizards' story (Grace, 1907a) was provided
to all informants from Ngāti Koata and Ngāti Kuia. Interview participants self-selected and/or were
recommended by participants and extended family members. Each session lasted between 0.5-2
hours and was attended by a research facilitator. All interviews were electronically recorded. Analysis
of interview material was inductive and consisted of (i) 'content analysis' whereby ideas or words
were identified along with the frequency of their use, (ii) 'thematic analysis' whereby the principal
elements emerging from the data were examined and sorted, and (iii) cross-checking the integrity of
emergent ideas and interpretations through follow-up discussions with key informants with
adjustments made where necessary. Central to these analyses was an emphasis on participant views
about the narrative (rather than the meaning the researchers brought to the research). Secondary
sources of information provided supplemental support. In following such methods, we sought to
avoid subjecting the story to external judgements, or in other words, risk turning the story into
something it is not.
**5. THE RIVAL WIZARDS (ABRDIGED)**
An abridged version of the Rival Wizards story is outlined below to provide context for the
summarised commentaries that follow. Importantly, in abridging the story, we are mindful that where
one chooses to begin and end a story can alter its shape and meaning, and so we encourage a reading
of the full story as published by Grace (1907a).
**5.1 Synopsis of the story**
The story begins with Rongomai, a "wizard-chief" renowned for being able to shape-shift from
monstrous to human form. One day, with his revered greenstone fish-hook (named Huakai after one
of his most famous ancestors) Rongomai paddled from his island settlement of Motiti to the shore of
the mainland opposite the settlement of Motu to fish for *hapuku* [wreckfish] and *kahawai* [A-NZ



salmon]. Boastful of his prowess as a fisherman Rongomai soon lost Huakai to a large fish, leaving him
miserable and despairing. Te Pou, the rival "wizard-chief" from Motu, watched these proceedings
from the shore. Famed also for his shapeshifting capabilities, Te Pou waited until after dark and then
stepped into the water turning himself into a shark and searched for the coveted hook. However,
Rongomai initiated an immense fishing haul, and relocated 'Huakai'; although there was
consternation at a large hole in one of his nets presumably caused by a shark. Te Pou was furious at
Rongomai for having found 'Huakai', and for almost having been caught in his fishing nets. Vowing
revenge, Te Pou later swam to the village of Motiti and in the middle of night he thrust a burning stick
into the thatch of Rongomai's house. Rongomai's human form was burnt and he was thereafter
confined to an aquatic existence as a veracious and malevolent salmon. The fish from the coast near
Motu were soon thereafter driven away by Rongomai, and then while swimming, Te Pou's son,
Kopara, was eaten by Rongomai. The mourning Te Pou subsequently planned a great farewell for his
son, but realising the scarcity of fish he transformed himself into a porpoise and travelled to have an
audience with Tangaroa, the supreme ruler of the sea. Here Te Pou requested that all the salmon
over whom Tangaroa held sway to come to Motu, be summoned to the mouth of the river, to weep
for his son. Tangaroa agreed to the request, but also indicated his interest in joining the occasion. In
reply Te Pou acknowledged the great pleasure this would bring, but he cautioned that the water at
Motu is hardly deep enough, with extensive mudflats and the river so shallow that it would be a most
inconvenient place for Tangaroa. Returning home Te Pou advised his people to prepare their nets for
the fish that would come, advising that he expected the pick of three fish for his own use. Standing on
the shore Te Pou proceeded to say incantations while Titipa, the next chief in command and secret
rival, ignored Te Pou's requests. When the great haul of fish was pulled ashore, Te Pou returned to
inspect the catch only to find Titipa claiming it. Te Pou therein warned all to stand back from the
beach as three great waves were called forth, advancing and receding from the beach, eventually
taking Titipa with them. The story ends with Te Pou selecting the three largest fish from the collective



haul, gifting the first to his son and the sea, the second to his wife, and the third for himself, ending
Rongomai's existence.

## 6. STORY-TELLING THROUGH WHAKAPAPA[3]

### 6.1 Narrative sources

The published version of the 'Rival Wizards' story (Grace, 1907a) was "not known" by the informants
from Ngāti Koata and Ngāti Kuia prior to the formal discussions carried out for this study. There were,
however, many repeated qualifications about parts of the narrative being very familiar. Independent
of one another, informants from both kin groups initially expressed "I am not familiar with the story",
"the story does not ring a bell for me", "I've never heard our people talk about it" and, among others
"the first time you gave me the story is the first time I had come across this". There was, however,
widespread awareness of Karepa Te Whetu (the informant of the story), first by the research
participants from Ngāti Koata who hold direct genealogical connections, and second by those from
Ngāti Kuia who recognised his name from pan-tribal history. From these collective voices, we know
that Karepa Te Whetu lived on D'Urville Island (Rangitoto) and that he was the elder son of Te Whetu,
a respected Ngāti Koata leader who migrated with other Ngāti Koata descendants from the North
Island in the 1820s to settle on Rangitoto and other areas across the northern South Island (Figure 1).
Te Whetu had a settlement at Te Marua (north-eastern side of Rangitoto), which is known for its
swampy ground and cliffs. An informant suggested that Karepa Te Whetu most likely grew up at Te
Marua alongside kin from Ngāti Koata and the already occupying people of Ngāti Kuia. For example,
an informant from Ngāti Koata reflected: "Ngāti Koata moved down here in the 1820s. And there was
a whole big history on that island [Rangitoto] before we moved in so I wonder how much of that
history, those stories, that he [Karepa Te Whetu] heard". In his later years, it was widely understood
that Karepa moved to Croiselles Harbour where he spent his final days (although one informant

---

[3] Ancestral and kinship linkages to people and place, genealogy, literally means 'to place in layers'.





suggested that he may also have lived at Taranaki for a while). According to Grace (1907b) it was
during this period that he got to know Karepa Te Whetu, leading eventually to the sharing of
numerous stories, until Karepa's death in 1903.
Reflecting further upon the 'Rival Wizards' story shared by Karepa Te Whetu with Alfred Grace, many
informants from Ngāti Koata and Ngāti Kuia noted that knowledge holders had probably passed on
and/or moved away from the Island, thereby taking many of their stories with them. One informant
also remarked that, "Some of our old people were cautious about who they told things to, so they
never told them". Other explanations for not knowing the 'Rival Wizards' story included reference to
changes in the resident population of Rangitoto following the arrival of the first Ngāti Koata peoples
and thereafter the broader social-cultural changes stemming from the arrival of the first missionaries.
Statements from both Ngāti Koata and Ngāti Kuia informants included: "What happened prior to the
*heke* [migration] … there are a lot that probably won't know what those stories were … so yeah it is
probably a Ngāti Kuia story", and "These events [in the story] are before Ngāti Koata. It's probably a
Ngāti Kuia story eh?" and "Ngāti Kuia lived on the Island, right up until the 1870s, early 1880s. My
great grandfather was born on the island [Rangitoto] but he was straight Kuia... And then all the Kuia
left… so lots of those *korero* [stories] about Rangitoto were not spoken about anymore. Ngāti Kuia lost
a lot of those *korero* whereas our Ngāti Koata-Kuia relations who stayed on the island retained their
knowledge of the place". Whatever the case might be, two informants (one from Ngāti Koata and the
other who recognised their links to both Ngāti Koata and Ngāti Kuia) also affirmed that they had no
reason to doubt the story from Karepa Te Whetu: "If it [the story] came from Karepa, I have no reason
to doubt it". Finally, upon questioning the informants about the role of Alfred Grace in the telling of
the story there was no mention of misgiving or distrust, as is common for other Māori when reflecting
on the work of other ethnographers of the time (Mikaere, 1995; Smith, 1999; Haami, 2012).
### 6.2 Key elements and story-telling devices
Many of the informants expressed familiarity with the places and contextual details described in
Grace's account. The most common reflections included reference to the two settlements named in





the story, Motiti and Motu. Initial discussions suggested informants were unaware of such settlement
names on, or surrounding, Rangitoto. However, several informants from Ngāti Koata and Ngāti Kuia
(in conversations independent of one another) were quick to point out that there is a Motuiti Island,
also known as Moutiti, Motiti and Victory Island, just off the northern coast of Rangitoto (Figure 1).
For example, one Ngāti Kuia informant stated: "In the old books, it is referred to as Motiti and
Moutiti. Motiti - that could be just a misspelling if it has been orally translated. That kind of thing was
prevalent when they [ethnographers] were transcribing as they heard it and I would expect it would
have been the same kind of situation here…Motiti, Moutiti, Motuiti". However, one Ngāti Koata
informant questioned these possible linkages, drawing specific attention to there being no beaches
on Motuiti and no visible signs of having been occupied (i.e. pits or middens). Notwithstanding these
literal inconsistencies, the same informant described the island as an important site for ongoing
traditional harvesting of wild-foods.
With reference to the settlement of Motu, one Ngāti Kuia informant noted the proximity of Motuiti
Island to the historical settlement at Otu Bay at the northern end of Rangitoto, and questioned
whether Otu Bay might be a misspelling of Motu (Figure 1). Another Ngāti Kuia informant questioned
whether Motu might be a shortening of a longer name such as Motungararara (now formally named
Titi Island) which was not only the site of a settlement held by Te Pou Whakarewarewa [an historical
figure understood to have lived during the late 18[th] century] but also a position where he had control
of all the area. It was surmised by another informant from Ngāti Koata that by using the name Motu
(translates as Island) Karepa Te Whetu may have been 'generically' referring to all the islands in the
area, not just a specific place. Alternatively, another informant from Ngāti Koata offered that "just
because people don't know this name 'motu' it doesn't mean that there wasn't a place called motu,
but the name may have been buried or usurped by new peoples coming in…". Given these initial
commentaries, there was general agreement that the story was derived from (and/or around)
Rangitoto but it was not possible to confirm any specific location.

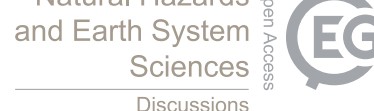

The description of extensive mudflats and a shallow river at the settlement of Motu, also led some
informants to specifically reflect on several locations on Rangitoto and its surrounds with similar
physical characteristics. For example, a Ngāti Koata informant stated "When I think about that, I think
about Whangarae on the Nelson mainland, just before Okiwi Bay. It was closer than other places on
the Island. My recollection is going there as a child for a *tangi* [funeral] and we anchored our boat out
there and on the low tide it was stranded. We just waited for the tide to come back in again. And
there was a big settlement in that place…at Whangarae… That area is still owned by Ngāti Koata. Not
many people live there now but there are a lot of owners…you could class that as part of D'Urville
Island [Rangitoto]" (Figure 1). The same informant emphasised that these places were not regarded
as separate by the people living in these areas and that any attempts to locate places referred to in
the story need to understand that the sea connected all the islands and the mainland as well as the
settlements situated along their coasts. The informant added "there is another place on D'Urville
Island which is in the Manuhakapakapa Bay. The water there and particularly Opitiki Bay was heavily
populated pre-Ngāti Koata and probably even Ngāti Kuia…and the water there is shallow". In addition,
specific reference to the "river" at Motu led some informants to contemplate the absence of rivers on
the Island as well as the neighbouring mainland. While this was inexplicable for some, informants
from both Ngāti Koata and Kuia recounted that the extensive use of geomorphic names such as
'sounds' and 'arms' across the northern South Island today refer to locations that were traditionally
referred to as *awa* [river]. For example, "Te Hoiere – is a good example of that. Today we talk about
the Pelorus River and Pelorus Sound, as opposed to Te Hoiere being one big entity into the Cook
Strait. Even some of the place names through the sounds Awaiti and Awanui, they were calling arms
at the time also, so even if we were thinking about D'Urville Island and Port Hardy and Greville
Harbour and all of those places, there are lots and lots of little arms all over the place [that would
have had names]" (Figure 1). Such contextual nomenclature may thereby explain the use of the term
'river' in the story.



Ancestral protagonists were another common element discussed by all informants. However, it is
important to qualify that most key informants from Ngāti Koata either declared no knowledge of the
names or that the names (or at least some) pre-dated the arrival of Ngāti Koata people to the region.
In contrast, most of the key informants from Ngāti Kuia recognised the names of the central
protagonists, and quickly confirmed linkages, citing genealogical books and historical transcripts (e.g.
Meihana Whakapapa Book, no date; Hemi Whakapapa Book, no date), and the ongoing use of such
names today. As one respondent declared, "Rongomai, Te Pou and Titipa - I know all those names"
and another stated "Te Pou - yep that's my father's middle name. Te Pou is a very common name for
Ngāti Kuia. Every Peter is a Pou … so that name's a common one". Another said, "Te Pou and
Rongomai have been commemorated down to the present day by the repeated use of their names in
the lines of Ngāti Kuia *whānau* [families]". The sacred fishing hook 'Huakai' used by Rongomai was
recognised by another Ngāti Kuia informant as a term used by recent generations of Ngāti Kuia. It was
also noted that the ancestors named in the story also derived from quite different periods of time.
Thereby, any attempts to historicise elements within the story based on genealogy would more likely
than not result in looking for detail that is not there. Two commentaries summarise these sentiments:
"Such stories were not necessary told in a linear fashion" and "The stories don't follow linear ways of
telling a story and that is important because you can have different ancestors from different times to
celebrate those people, to remember them, to remember a lesson… so they are not forgotten". In
this way, it is the protagonists rather than chronological dimensions of time that are of most
relevance.
Other contextual aspects in the story considered highly relevant to locating the narrative included
familiarity with large sharks and *kahawai* (salmon) in the area, particularly at Manuhakapakapa
Harbour (Figure 1). For example, multiple references to *kahawai* were made by Ngāti Koata
informants who grew up on Rangitoto Island: "*Kahawai* is everywhere [around Motuiti Island] …we
catch *kahawai*, we get it quite easy…", "*Kahawai* were plentiful around the Island [Rangitoto]… like at
Kape [Manuhakapakapa Bay] … there was a big *kāinga* [settlement] here". And, "I can tell you a story.



We had my dad's uncle, and he was Ngāti Kuia. He was brought to live with us on the Island
[Rangitoto], and he didn't like the people he was staying with. This was at Ohana. So, he left for two
or three days and there was no sign of him. So, they sent back to his people in Okoha (in the Pelorus
Sound), and they asked are any fish there? Our people responded yes there is a lot of *kahawai* on the
Puna (Te Puna Bay) side of Ohana. They said that's where you will find him. What he used to do is dive
under the water and put his thumb and fingers into the gills of the kahawai and that's what he lived
on until they found him". Upon querying informants about which bay might represent the traditional
settlement of Motu named in the story, some considered the Manuhakapakapa Harbour as a possible
analogue, while others pointed out that Whangarae, Otu Bay as well as Skull Bay in Port Hardy are
equally possible given the significant settlements at all of these neighbouring places in the past.
Notwithstanding these reflections, many informants considered these contextual aspects in the story
highly relevant for connecting the story to the area. For example, one of the informants from Ngāti
Kuia stated: "It is not only the descriptive language of catastrophic waves being called ashore, but the
other details, that make us believe we are in the place".
References to the power of prayer and incantation [*karakia*] as well as shapeshifting [*turehu*] in the
story were also identified as highly relevant to any claims of the narrative coming from the northern
South Island. Ngāti Kuia informants emphasised not only this power, but also the reputation held by
the "tōhunga" [priest; expert in traditional lore; person skilled in specific activity; healer] of Ngāti Kuia
to modify the elements. For example, "We were known as *te iwi karakia* [the necromancing people]
…but not the kind that do *makutu* [dark incantations]. Our *karakia* were very much a demand, that
was the *mana* [authority, control, influence, prestige] and power of the *tōhunga* [priest; expert in
traditional lore; person skilled in specific activity; healer]. We are connected to all of our *Atua* [Gods,
deity] and we are made of our *Atua*". These discussions also led one of the informants from Ngāti Kuia
to reflect specifically on the significance of the incantation used in the story and whether the
description of destructive waves was due to a tsunami or a phenomenon manifest through
metaphysical forces. In response, the informant answered: "what I do know is that our people were




recognised as very strong *kaikarakia* [necromancers]". Mitchell and Mitchell (2004) have also pointed
out that Ngāti Kuia have long been recognised for their powers in this regard and historical transcripts
are known to contain *karakia* about how to control the sea and the waves, with many references to
Rangitoto (Smith, 1889). The story also incorporates multiple references to Te Pou and Rongomai
'shapeshifting' or transforming themselves into various life-forms from the sea, from whale and shark,
to porpoise and kahawai. Again, several informants from Ngāti Kuia affirmed a deep familiarity with
such details, including acceptance of the supernatural and the metaphysical world. For example,
"Shapeshifting, that is acceptable to me. I grew up with that *korero* [story]" and "Kaikaiawaro is our
*kaitiaki* [person, group, being that acts as a carer, guardian, protector and conserver] and he takes the
form of a dolphin". Further still, the familiarity with these elements in the story extended to
recognition among many of the Ngāti Kuia informants that they were descendants of Kaikaiawaro,
and that he is present in their genealogy as an ancestor rather than an Atua. As an informant
declared, "Yes...when I was reading this Te Pou goes to visit Tangaroa and when he transforms himself
it was like we know that because Kaikaiawaro who is in our *whakapapa* as a person who could
manifest himself as a dolphin… We are the descendants of Kaikaiawaro".

462       6.3 Memorials and analogue stories

Reflecting upon the specific narrative of Te Pou [the principal protagonist in the Rival Wizards story]
calling forth catastrophic waves, many informants from Ngāti Koata and Ngāti Kuia regarded this
account as most likely referencing direct experience with past tsunami inundation. Although, almost
all of these informants were quick to point out that they did not know where this story occurred
and/or when it happened, and that the narrative was being told within a framework of deities and
super-natural humans with influence over the elements. Consideration of the narrative as a tsunami
tradition also led several of the informants to note similarities with the destructive waves described in
another story from Moawhitu [Greville Harbour] on the western side of Rangitoto (Figure 1).
According to these commentaries a tsunami, possibly occurring in the 1400s or 1500s, drowned
nearly all people living around Greville Harbour, and their bodies now lie in the surrounding sand



dunes. For example, "Yes, there was a great big tidal wave. I heard it when I was a kid. My
grandmother told me when I was a child. This story is *tuturu tika* [genuinely truthful]. I don't question
it". The story of Moawhitu was also recounted by Karepa Te Whetu to Elsdon Best and published in
the Journal of the Polynesian Society in 1893 (Te Whetu, 1893). It describes the people of Ngai-
Tarapounamu who settled Rangitoto Island and a breach of *tapu* [sacrosanct, forbidden, inviolable] by
a local woman which led to the gods stirring up the deep ocean and causing great waves to sweep
away people where the woman was living. Phillipson (1995) purports that the "tidal wave" occurred
some-time in the sixteenth century, while Cope (2011), Chagué-Goff and Goff (2012a, 2012b) and
Cope et al., (2012) indicate the previous century as more likely based upon the inferred timing of a
Māori occupation layer beneath marine gravels at Moawhitu as well as palaeotsunami evidence from
neighbouring sites across region. Meanwhile, Mitchell and Mitchell (2004) referred to the "tidal
wave" as *Tapu-arero-utuutu* [vengeance for the breaking of strict food preparation practice] and
postulated that the people already living on the Island prior to the arrival of the kin-group Ngai-
Tarapounamu may have been from the ancient Waitaha peoples and/or early Ngāti Kuia lines. It is
also noteworthy that one informant familiar with the name Tapu-arero-utuutu identified a stand of
offshore rocks to the south west of Rangitoto by the same name (Figure 1). The association of this
name with tsunamis and its close location to Rangitoto however were not mentioned.
More than one informant questioned whether the Rival Wizards narrative might be a retelling of the
Moawhitu tradition. One informant questioned where knowledge of the Moawhitu tradition had
actually come from. For example, "I have heard the *korero* about Moawhitu and the tsunami there,
but I was told by my uncle (and he is passed away now) that the people were labouring men but also
avid readers so I cannot say whether that story was one that we had or what he had read and then
became ours". Meanwhile another informant reflected that the [Rival Wizards] story might not
necessarily be referring to Moawhitu, but rather the Manuhakapakapa area due to the strong
references to kahawai and the abundance of people in the area: "This certainly could have been a
place where that *korero* might have been had". In contrast, Otu Bay and Skull Bay were also identified





by other informants as equally likely sites referenced in the story. As noted earlier, one Ngāti Koata
informant reflected that the name motu might have not only been used in a general sense but also to
reflect that there are many places here that were likely affected by the extraordinary waves described
in the story and so a generic settlement name was used to capture this. Whatever the case may be, in
considering the specific sites and sources for the Rival Wizards story there was widespread agreement
(although not total) that the story and its elements derived from Rangitoto and the connected places
and peoples that surround the northern South Island. As one respondent noted, "It's definitely got
the feel that it comes from this place".
**7. MAORI ORAL HISTORIES AND NATURAL HAZARDS SCIENCE**
**7.1 Lessons and opportunities**
By engaging directly with informants from Ngāti Koata and Ngāti Kuia it is evident that there is a deep
familiarity with the different elements contained in the Rival Wizards story. This includes knowledge
of past tsunami impacts on, and surrounding, the island of Rangitoto. Dialogue may not have included
familiarity with the specific story itself, but ancestral relationships were confirmed between
informants of Ngāti Koata descent and the original informant of the story Karepa Te Whetu as well as
those informants of Ngāti Kuia descent and the leading protagonists in the story. Many other aspects
of the story are also deeply rooted in the enduring knowledge of Māori histories across the northern
South Island. And, while the exact location of catastrophic waves could not be confirmed, most of the
informants (from both Ngāti Koata and Ngāti Kuia) regarded the story as incorporating direct
experience with past tsunami inundation(s) on Rangitoto Island and the neighbouring coastal
surrounds.
More broadly, this work confirms that Māori oral histories are dynamic, even when committed to
writing in an ethnographical text. The Rival Wizards story holds multiple purposes comprising
elements of culture, place, identity, lineage, history and in this case, environmental risk. It is also clear
that ancestral and kinship linkages to people and place (i.e. *whakapapa*) are central to the





construction and ongoing retelling of Māori histories. Royal (1992: 21) affirmed this notion stating
that *whakapapa* is "the fabric upon which tribal histories sit" generating meaning for human
behaviours and understanding in the Māori tribal world. Further, Roberts (2012) explained that
*whakapapa* is used in story-telling as a construct for mapping the natural world and its phenomena;
thereby acting as a "mental map" of place. And most recently, Kelly (2016) has reflected that Māori
knowledge was stored layer by layer, referencing places, ancestors and the actions of protagonists
as 'memory cues' to retain vitally important information. The specific layering of contextual detail
in the Rival Wizards story affirms these connections and relationships between the natural and
metaphysical worlds, including the narrative structures critical to cultural endurance and memory.
Our working with informants from Ngāti Koata and Ngāti Kuia also highlights that Māori oral histories
can complicate scientific definitions of what constitutes events. That is, the earth sciences typically
treat events as discrete and bounded but in the case of the Rival Wizards a different paradigm with
non-linear contextual details is used to establish layers of meaning with ancestral protagonists from
different epochs of genealogical time. Tau (1999) reflects that events in the Māori world are often
recalled relative to known ancestors rather than fixed at some objective point in time. Further he
points out that trying to apply chronology to genealogical time is akin to historicising a past that was
not intended to constitute a linear history. In short, Mātauranga Māori orders itself differently, and
thereby the risk of misinterpretation is high when stories and their elements are not understood
within the context of ancestry and cultural experience (Roberts et al., 1995; Berkes, 1998; King and
Goff, 2010).
The methodology underpinning this research provides an example of how the earth system sciences
as well as the knowledge-practice-belief complex of Mātauranga Māori can benefit from engaging
collaboratively with one another. Confirmation of deep connections to the Rival Wizards story and
subsequent affirmation of ancestral experience with past tsunami(s) across the northern South Island,
casts off earlier assumptions that the story might derive from the eastern Bay of Plenty (King and
Goff, 2010). Further, this study emphasizes the value of such engagements, particularly for scientific



researchers who seek to learn from the historical experience captured in Māori oral histories. From
this epistemological position, we agree with Styres (2008) who argued that the challenge for
researchers from the academy of science is to go beyond traditional methodological approaches and
assumptions about research which select and frame stories from the point of view of the dominant
culture. Further, we concur with Johnson et al. (2016: 3) that a reframing of science is needed
whereby "one is drawn to the wider value of a dialogue across knowledge systems that is humble,
respectful and hopeful; which recognizes not only the need to acquire knowledge, but also the need
to transform and respond to different knowledges, understandings, meanings, and opportunity".
Although, we simultaneously acknowledge that this is deeply challenging because the research
structures around us constantly push and pull us to neglect and compromise these values, ethics and
practices. Further, we recognise that research framing will not solve all the problems associated with
the hierarchies of power and knowledge production (Mustonen, 2014).
Notwithstanding these ongoing tensions, engaging in this work can help to promote "plural spaces" of
learning that contribute to the reclaiming of stories and culture as well as the development of new
knowledge and new questions (Howitt and Suchet-Pearson, 2003; Zanotti and Palomino-Schalsha,
2006). For example, the work undertaken in this study contributes to a number of projects currently
being undertaken by Ngāti Koata and Ngāti Kuia by adding to their existing stores of knowledge. This
research space also provides an opportunity for the knowledge-practice-belief complex of
Mātauranga Māori to engage with the academy of science about tsunami disturbance, recurrence
and risk. And, as already articulated, there remain many unrealised opportunities for Mātauranga
Māori to inform the earth system sciences about extreme hazard episodes and risk along the A/NZ
coastline over the past 1000 years (King and Goff, 2010; King, 2015; King et al., 2017). Such work
however will require greater attentiveness to relationships among people involved in the research,
including the need to be aware of contemporary developments political, epistemological and
methodological practice.
**8. CONCLUSIONS**




This work confirms northern South Island Māori links to 'The Rival Wizards' narrative, including
knowledge of ancestral experience with a past tsunami, possibly even multiple events, on, and
surrounding, Rangitoto (D'Urville Island). While we cannot confirm the exact location of the story, it is
evident from the multiple exchanges with key informants from Ngāti Koata and Ngāti Kuia, that the
narrative comprises multiple layers of history and meaning. However, notwithstanding these
confirmations, to engage with oral histories (and the people who link genealogically to such stories)
requires close attention to a politics of representation, which includes considerations about how
knowledge is constructed, applied and legitimised. It also demands sensitivities to the production of
'new' and 'plural' knowledge itself. Individuals and families from Ngāti Koata and Ngāti Kuia have
permitted us to record some of their history, because they share the view that there are multiple
benefits to be gained by learning from differences in knowledge, practice and belief. Further still, the
'retelling' of this narrative offers an opportunity to relive ancestral experience across different epochs
of genealogical time. The account offered in this paper makes these narratives available to a new
audience (including those families who no longer have access) and recites these in ways that might
encourage those more intimately connected to know and transmit these oral histories differently.





**ACKNOWLEDGMENTS**
The authors acknowledge the key informants from Ngāti Koata and Ngāti Kuia without whom this
work could not have been undertaken. The research was funded by the Resilience National Science
Challenge - Vision Mātauranga (Grant Agreement. 28378) and the NIWA Strategic Science Investment
Fund - Hazards, Climate and Māori Society (Grant Agreement. C01X1702). Dr Mere Roberts and Dr
Bruce McFadgen are thanked for their constructive review comments. Stephanie Huriana Martin is
thanked for her assistance with Te Reo Māori.





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



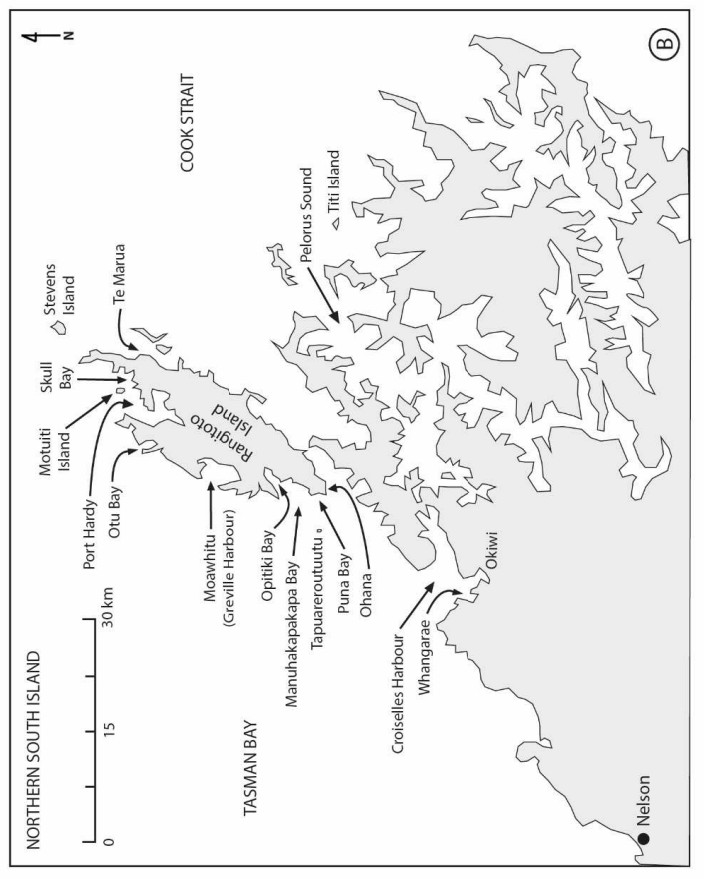

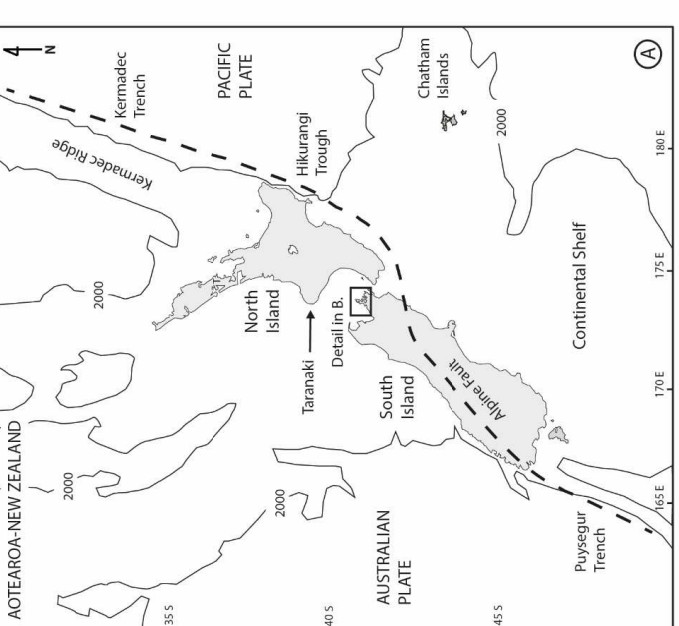

Figure 1: **(A)** Aotearoa-New Zealand's tectonic location in the South Pacific showing the Australian-Pacific plate boundary as a dashed line. The submerged continental shelf boundary is loosely defined by the 2000 m isobaths (adapted from Carter et al. (1988)). **(B)** Rangitoto Island (D'Urville Island) and surrounding locations mentioned in the text.