# Peer review of "Māori oral histories and the impact of tsunamis in Aotearoa-New Zealand"

_Natural Hazards and Earth System Sciences, 2017_

## Referee Comment (RC1) · M. Crozier (Referee) · 12 Dec 2017

Maori oral histories and the recurring impact of Tsunamis in Aotearoa- New Zealand.

Given that this manuscript has been submitted to NHESS, one can assume that the aim was to add to our understanding of a hazard, in this case tsunami. In New Zealand, where the historic / written record is so short, the opportunity to extend the information base by exploring orally transmitted stories of the pre-European Maori is certainly worth investigating. The authors thus found an appropriate story that contained reference to three catastrophic waves (the story was written-down by Grace (1907) from a conversation with Karepa te Whetu, who lived for sometime in the north of the South Island). They then asked members of two Iwi with residential history in the north of the

[Figure]

South Island, essentially 'was it their story?!' First, none had heard the story, nor could any specific location of the three big waves be unequivocally determined. However, the original Maori source used by Grace, Kerepa te Whetu, was known of by some of the respondents of one of the Iwi, and they were also familiar with people's names used in the story. After sifting through the respondents' comments and dealing with apparent contradictions by resorting to a number of reasonable devices such as identifying miss-spellings, different concepts of what constitutes a place, and changes of meaning (e.g. 'sound's and 'arms of the sea' could conceivably represent the rivers referred to in the story) the authors considered that they had the general location of the story right. However, I must say that using the presence Kahawai and sharks to point to the proposed location in the story was stretching credibility, as they are abundant in many widespread parts of New Zealand; and I would have expected critical comment on this aspect. Convinced that they had the general area of NZ correct, if not the specific locality of the big waves, it followed that they must be talking to the right Iwi. So therefore what did we learn about Tsunami from this story? If the three big waves in the story were in fact a reference to a tsunami (rather than a literary devices, or representation of some super natural force, or a physical manifestation of an emotion such as revenge) what information did we gain from this form of discourse analysis. The least we could hope for is some understanding of magnitude, date and location of the assumed tsunami. The study could not convincingly provide this or indeed much else about a proposed paleo-tsunami (thus I believe the first sentence of the abstract greatly overstates what the study revealed about the ancestral experience with tsunamis). So is the study worth recounting? The answer is yes, for the following reasons. This paper is not really about hazards and Tsunami. Rather, it is about a methodology for cross-cultural, cross-temporal investigation. It is about exploring and relating two different approaches to understanding the world in both the human and natural settings. In this sense it makes an excellent well-written contribution to our pursuit of knowledge. The study presented here shows a very sensitive and thorough approach to investigating a record that is different from the ones normally resorted to by modern western sci-

ence. It outlines the pitfalls of working without an understanding of epistemology. On the whole, the claims and 'confirmations' are treated with adequate caveats and the authors are acutely aware of the mistakes that can be made by not fully understanding the purpose and power of the narrative and the disposition of the narrator. This paper will provide useful guidance to future investigators of pre-European oral histories irrespective of whether credibility can be ascribed to this story as account of a Tsunami.

Whether the paper would have more impact and value in a journal devoted to the philosophy of science; or indeed cultural studies or social anthropology is a question for the editor.

Note: Incidentally, there does not appear to be any reference to recurring impact of tsunamis in the study – therefore the title should be modified. Perhaps the methodological value of this paper could also be reflected in the title. Note: suggest reformulating the abstract to reflect better the aim, method, findings and principal contribution made by your study (see comment in papa 2 above). Michael Crozier 12 December 2017

---

## Referee Comment (RC2) · W. Dudley (Referee) · 12 Dec 2017

This is extremely important work. Indigenous populations have had their own experiences with natural hazards and many have developed techniques for coping in traditional ways. Western science can learn a great deal from indigenous knowledge, furthermore combining the two helps emergency managers more effectively educate local populations about the best way to prepare, respond, and cope with natural hazards, thus bridging the gap between western science and indigenous knowledge and thereby creating a more effect synergistic relationship.

---

## Author Comment (AC1) · 24 Jan 2018

(1) Referee comment #2: "This is extremely important work. Indigenous populations have had their own experiences with natural hazards and many have developed techniques for coping in traditional ways. Western science can learn a great deal from indigenous knowledge, furthermore combining the two helps emergency managers more effectively educate local populations about the best way to prepare, respond, and cope with natural hazards, thus bridging the gap between western science and indigenous knowledge and thereby creating a more effect synergistic relationship."

(2) Author's response #2: We are very grateful for the reviewer's endorsement of this research work.

[Figure]

(3) Author's changes in manuscript #2: No changes are required to the manuscript.
* * *

---

## Author Response (AR1)

DN King

PO Box 109-695

NIWA Ltd, Auckland

New Zealand

Ph: + 64-9-375 2086

Fax: + 64-9-375 2051

E-mail: darren.king@niwa.co.nz

Editorial Office

Natural Hazards and Earth System Sciences

7th February 2018,

Dear Dr. Glade (Editor),

Please find below all responses and actions to the points raised by two reviewers for the manuscript

"Māori oral histories and the impact of tsunamis in Aotearoa-New Zealand'. In addition, as requested, the 'revised' manuscript is provided showing all final marked changes. The authors are grateful for the opportunity to improve this manuscript.

Response to Referee #1 - Michael Crozier.

(1) **Referee comment #1A:** "Given that this manuscript has been submitted to NHESS, one can assume that the aim was to add to our understanding of a hazard, in this case tsunami. In New Zealand, where the historic / written record is so short, the opportunity to extend the information base by exploring orally transmitted stories of the pre-European Māori is certainly worth investigating. The authors thus found an appropriate story that contained reference to three catastrophic waves (the story was written-down by Grace (1907) from a conversation with Karepa te Whetu, who lived for some time in the north of the South Island). They then asked members of two Iwi with residential history in the north of the South Island, essentially 'was it their story?!' First, none had heard the story, nor could any specific location of the three big waves be unequivocally determined. However, the original Māori source used by Grace, Kerepa te Whetu, was known of by some of the respondents of one of the Iwi, and they were also familiar with people's names used in the story. After sifting through the respondents' comments and dealing with apparent contradictions by resorting to a number of reasonable devices such as identifying miss-spellings, different concepts of what constitutes a place, and changes of meaning (e.g. 'sound's and 'arms of the sea' could conceivably represent the rivers referred to in the story) the authors considered that they had the general location of the story right. I

must say that using the presence Kahawai and sharks to point to the proposed location in the story was stretching credibility, as they are abundant in many widespread parts of New Zealand; and I would have expected critical comment on this aspect."

(2) **Author's response #1A:** The referee provides a very thoughtful account of the logic of the manuscript, and is correct in his assumption that the aim/thesis of the research is to add to scientific and Māori narratives about tsunami hazard (and history) across the northern South Island of

Aotearoa-New Zealand. The secondary aim of the manuscript (identified later in his commentary) is to demonstrate the need for closer attention by the geoscience community to epistemological, political and methodological issues when exploring the benefits of differences in Māori knowledge (and by inference Indigenous Knowledge) and science about tsunamis. The referee also helpfully points out the need for a critical comment about the ubiquitous nature of kahawai and sharks around the A-NZ

coast. Notwithstanding our agreement about the added value that such a sentence would make, it is important to make clear that the authors do not actually confirm the specific location of the story based on the presence of kahawai and sharks. Rather we argue that the key elements in the story (which includes close relationships with kahawai and sharks in areas where there were previously large settlements) provide strong collective evidence for connecting the story to the Rangitoto (D'Urville Island) area, not a specific place on the Island.

(3) **Author's changes in manuscript #1A:** The authors consider that editing the abstract to more clearly reflect the principal objectives and outcomes of the work will address any uncertainty about the aims of the research. In addition, a new sentence will be added to the relevant paragraph within '6.2 – Key elements and story-telling devices' to provide a critical comment about the ubiquitous nature of kahawai and sharks in A-NZ waters; including acknowledgment that such information alone is insufficient to draw any critical conclusions about a proposed location for the story. Added to this, in order to remove any remaining ambiguity an extra sentence will be added to the manuscript to confirm that it is the collective evidence from multiple informants that connects the story to

Rangitoto (D'Urville Island).

All changes have been made. Please see the amended Abstract, section 6.2 and the Conclusion.

--------------- (1) **Referee comment #1B:** "Convinced that they had the general area of NZ correct, if not the specific locality of the big waves, it followed that they must be talking to the right Iwi. So therefore, what did we learn about Tsunami from this story? If the three big waves in the story were in fact a reference to a tsunami (rather than a literary device, or representation of some super natural force, or a physical manifestation of an emotion such as revenge) what information did we gain from this form of discourse analysis. The least we could hope for is some understanding of magnitude, date and location of the assumed tsunami. The study could not convincingly provide this or indeed much else about a proposed paleo-tsunami (thus I believe the first sentence of the abstract greatly overstates what the study revealed about the ancestral experience with tsunamis)."

(2) **Author's response #1B:** The authors understand that this general comment reflects a desire for new information from Māori Knowledge that would help shed light on tsunami magnitude, date and location; however, framing such preferences as "the least we can hope for" underscores a preference for certain kinds of data that sometimes simply are not part of, or important, within the 'knowledge'

complex that is Matauranga Māori. Notwithstanding this lack of "data" and respecting the reviewer's point about not overstating what the study can reveal about ancestral experience with pre-written tsunamis on Rangitoto (D'Urville Island), we consider that the presentation of the collective narrative in this work provides layers of place-based experience that relate in the words of the 'home-people'

at least one, if not multiple, encounters with pre-written tsunami on Rangitoto (D'Urville Island). Such a confirmation is a step towards not only plural knowledge co-existence but also plural knowledge development.

(3) **Author's changes in manuscript #1B:** The opening sentence of the abstract will be modified to reflect experience with at least one pre-written tsunami event on Rangitoto (D'Urville Island).

All changes have been made. Please see the amended Abstract.

--------------- (1) **Referee comment #1C:** "So is the study worth recounting? The answer is yes, for the following reasons. This paper is not really about hazards and Tsunami. Rather, it is about a methodology for cross-cultural, cross-temporal investigation. It is about exploring and relating two different approaches to understanding the world in both the human and natural settings. In this sense it makes an excellent well-written contribution to our pursuit of knowledge. The study presented here shows a very sensitive and thorough approach to investigating a record that is different from the ones normally resorted to by modern western science. It outlines the pitfalls of working without an understanding of epistemology. On the whole, the claims and 'confirmations' are treated with adequate caveats and the authors are acutely aware of the mistakes that can be made by not fully understanding the purpose and power of the narrative and the disposition of the narrator. This paper will provide useful guidance to future investigators of pre-European oral histories irrespective of whether credibility can be ascribed to this story as account of a Tsunami."

(2) **Author's response #1C:** The authors are grateful for the reviewer's close examination and endorsement of the methodological benefits of this research work. They have forced us to check our own assumptions about providing sufficient detail. Notwithstanding this, as explained above the aim/thesis of the research is to also add to scientific and Māori narratives about tsunami hazard (and history) across the northern South Island of Aotearoa-New Zealand. The authors thereby consider that editing the abstract to more clearly reflect these dual objectives and outcomes will address any potential uncertainty by future readers about the aims of the research.

(3) **Author's changes in manuscript #1C:** The authors consider that editing the abstract and the conclusions to more clearly reflect principal objectives and outcomes of the work will address any uncertainty about the aims of the research (this includes signaling planned work ahead to search for any remaining physical evidence of tsunami inundations on Rangitoto (D'Urville Island)).

All changes have been made. Please see the amended Abstract.
* * *
(1) **Referee comment #1D:** "Whether the paper would have more impact and value in a journal devoted to the philosophy of science; or indeed cultural studies or social anthropology is a question for the editor."

(2) **Author's response #1D:** This work is pitched at the natural hazards and earth system sciences research community and highlights not only the value and benefits of epistemological and empirical differences in knowledge about tsunamis, but also the increasing requirement for a broader set of skills among the geosciences. To place this work elsewhere would limit its impact across the

Geosciences.

(3) **Author's changes in manuscript #1D:** No changes are required to the manuscript.

Not applicable.
* * *
(1) **Referee comment #1E:** "Incidentally, there does not appear to be any reference to recurring impact of tsunamis in the study – therefore the title should be modified. Perhaps the methodological value of this paper could also be reflected in the title."

(2) **Author's response #1E:** The use of the word 'recurring' in the manuscript title is a general acknowledgement that there are multiple Māori oral histories from across A-NZ that record ancestral experiences with pre-written tsunami impacts (e.g. King and Goff, 2010 – NHESS). Notwithstanding this, if this confuses potential readers we agree that it should be removed from the manuscript title.

With respect to incorporating a message about the methodological value of the paper in the manuscript title, the authors maintain that promoting plural knowledge learning and development about tsunamis from respectful and humble encounters between different knowledge paradigms is of paramount importance.

(3) **Author's changes in manuscript #1E:** The word 'recurring' will be removed from the manuscript title.

All changes have been made.

-------------- (1) **Referee comment #1F:** "Note: suggest reformulating the abstract to reflect better the aim, method, findings and principal contribution made by your study (see comment in papa 2 above)."

(2) **Author's response #1F:** The authors acknowledge the reviewer's helpful suggestion here and agree that further work on the Abstract would help to give greater account of the principle objectives, methods, and findings of this research.

(3) **Author's changes in manuscript #1F:** Modifications will be made to the Abstract.

All changes have been made. Please see the amended Abstract.

--------------

Response to Referee #2 - Walter Dudley.

(1) **Referee comment #2A**: "This is extremely important work. Indigenous populations have had their own experiences with natural hazards and many have developed techniques for coping in traditional ways. Western science can learn a great deal from indigenous knowledge, furthermore combining the two helps emergency managers more effectively educate local populations about the best way to prepare, respond, and cope with natural hazards, thus bridging the gap between western science and indigenous knowledge and thereby creating a more effect synergistic relationship."

(2) **Author's response #2A:** We are grateful for the reviewer's endorsement of this research work.

(3) **Author's changes in manuscript #2A**: No changes are required to the manuscript.

Not applicable.

--------------------------------------------------------

In summary, on behalf of the contributing authors, I would again like to thank Natural Hazards and Earth

System Sciences for the opportunity to improve this manuscript. I would also like to acknowledge the very thorough and constructive comments provided by the reviewers.

Yours faithfully,

Darren Ngaru King

[revised manuscript text omitted]